# Intermolecular Vibration Energy Transfer Process in Two CL-20-Based Cocrystals Theoretically Revealed by Two-Dimensional Infrared Spectra

**DOI:** 10.3390/molecules27072153

**Published:** 2022-03-26

**Authors:** Hai-Chao Ren, Lin-Xiang Ji, Tu-Nan Chen, Xian-Zhen Jia, Rui-Peng Liu, Xiu-Qing Zhang, Dong-Qing Wei, Xiao-Feng Wang, Guang-Fu Ji

**Affiliations:** 1Xi’an Modern Chemistry Research Institute, Xi’an 710065, China; haicren@126.com (H.-C.R.); jiaxz1027@163.com (X.-Z.J.); news_arrival@163.com (R.-P.L.); 2Department of Physics and Engineering Physics, University of Saskatchewan, Saskatoon, SK S7N 5E2, Canada; lij505@mail.usask.ca; 3The Southwest Hospital of AMU, Army Medical University, Chongqing 400038, China; ctn@tmmu.edu.cn; 4School of Science, North University of China, Taiyuan 030051, China; scdxzxq@163.com; 5College of Food Science and Engineering, Henan University of Technology, Zhengzhou 450001, China; dqwei@sjtu.edu.cn; 6College of Life Science and Biotechnology, Shanghai Jiao Tong University, Shanghai 200240, China; 7National Key Laboratory for Shock Wave and Detonation Physics Research, Institute of Fluid Physics, Chinese Academy of Engineering Physics, Mianyang 621999, China

**Keywords:** two-dimensional infrared spectra, vibration energy transfer, cocrystal TNT/CL-20, cocrystal HMX/CL-20, non-covalent interaction, Mayer bond order density, impact sensitivity

## Abstract

Inspired by the recent cocrystallization and theory of energetic materials, we theoretically investigated the intermolecular vibrational energy transfer process and the non-covalent intermolecular interactions between explosive compounds. The intermolecular interactions between 2,4,6-trinitrotoluene (TNT) and 2,4,6,8,10,12-hexanitro-2,4,6,8,10,12-hexaazaisowurtzitane (CL-20) and between 1,3,5,7-tetranitro-1,3,5,7-tetrazocane (HMX) and CL-20 were studied using calculated two-dimensional infrared (2D IR) spectra and the independent gradient model based on the Hirshfeld partition (IGMH) method, respectively. Based on the comparison of the theoretical infrared spectra and optimized geometries with experimental results, the theoretical models can effectively reproduce the experimental geometries. By analyzing cross-peaks in the 2D IR spectra of TNT/CL-20, the intermolecular vibrational energy transfer process between TNT and CL-20 was calculated, and the conclusion was made that the vibrational energy transfer process between CL-20 and TNTII (TNTIII) is relatively slower than between CL-20 and TNTI. As the vibration energy transfer is the bridge of the intermolecular interactions, the weak intermolecular interactions were visualized using the IGMH method, and the results demonstrate that the intermolecular non-covalent interactions of TNT/CL-20 include van der Waals (vdW) interactions and hydrogen bonds, while the intermolecular non-covalent interactions of HMX/CL-20 are mainly comprised of vdW interactions. Further, we determined that the intermolecular interaction can stabilize the trigger bond in TNT/CL-20 and HMX/CL-20 based on Mayer bond order density, and stronger intermolecular interactions generally indicate lower impact sensitivity of energetic materials. We believe that the results obtained in this work are important for a better understanding of the cocrystal mechanism and its application in the field of energetic materials.

## 1. Introduction

High-energy and low-sensitivity energetic materials have always been the research target in the field of energetic materials [1,2,3]. However, high performance and safety are somewhat mutually exclusive for current pure compound explosives, which severely restrict their development and applications [4]. For example, 2,4,6,8,10,12-hexanitro-2,4,6,8,10,12-hexaazaisowurtzitane (CL-20, also known as HNIW) [5,6], first synthesized by Nielsen in 1987, is the most powerful commercially available explosive for practical application at present, which exhibits high density and high detonation velocity but with relatively high sensitivity to heat, friction and shock. In contrast with CL-20, 2,4,6-trinitrotoluene (TNT) [7,8] and 1,3,5,7-tetranitro-1,3,5,7-tetrazocane (HMX) [9,10] have low sensitivity to external impact and low economical production cost, but demonstrate poor density, low oxygen balance and modest detonation velocity. Nowadays, as an effective method to balance sensitivity and safety, cocrystal explosive technology is being extensively developed [11,12]. Botlon et al. prepared a TNT/CL-20 cocrystal with a molar ratio of 1:1, which showed closed detonation performance and oxygen balance close to CL-20 and exhibited lower sensitivity than CL-20 [13]. Then they also synthesized a new HMX/Cl-20 cocrystal explosive with a molar ratio of 1:2, which showed a detonation velocity higher than pure HMX but sensitivity values nearly the same as HMX [14]. Subsequently, there were many reports characterizing the TNT/CL-20 and HMX/CL-20 cocrystals via experimental and theoretical methods. Yang et al. prepared the TNT/CL-20 cocrystal by the cocrystallization in solution method and characterized the structure using single X-ray diffraction and infrared spectroscopy. They found that the CL-20/TNT cocrystal had lower impact sensitivity, and the impact sensitivity of CL-20 was obviously reduced by 87% [15]. By employing the mechanical ball-milling method, Hu et al. prepared the nano-TNT/CL-20 cocrystal and found that it had better thermal stability and safety through impact sensitivity tests [16]. The spray-drying method was used to prepare HMX/CL-20 cocrystal, and the impact and friction sensitivity of the cocrystal was tested and analyzed, which demonstrated that the cocrystals exhibited significantly reduced mechanical sensitivity compared with raw HMX [17]. Ghost et al. developed a new preparation method for HMX/CL-20 cocrystals and characterized the structure via spectroscopy, thermoanalytical tools, X-ray diffraction and microscopic techniques [18]. Hübner et al. characterized a nanoscale HMX/CL-20 cocrystal prepared by spray flash evaporation through tip-enhanced Raman spectroscopy and proposed that the HMX surface finishing might contribute to the impact sensitivity of HMX/CL-20 [19]. The major part of the theoretical approaches consisted of classical molecular dynamic simulations of intermolecular interactions, vibrational spectra and detonation performance, where the atomic interactions were based on sophisticated force fields [20,21,22,23,24]. All these works demonstrated that energetic cocrystals created a distinct solid-state arrangement at the molecular level and the intermolecular non-covalent interactions were the main driving forces for the cocrystal formation [25,26,27]. Furthermore, intermolecular hydrogen bonding, such as N-O…H, might alter the impact sensitivity by changing the rate of mechanical energy transfer from the shock front to vibrational degrees of freedom coupled to the reaction [28].

Since nitro (NO_2_) groups are ubiquitous chemical entities of energetic materials, the loss of which is an important step in the shock initiation of CL-20, HMX and TNT [29,30], the rate of energy transfer in the presence and absence of these weak intermolecular bonds is of interest to the greater explosive community. Ostrander et al. observed vibrational energy transfer between coherently delocalized states of a pentaerythritol tetranitrate (PETN) thin film using an ultrafast 2-dimensional infrared (2D IR) spectra and revealed that the vibrational lifetime of the NO_2_ asymmetrical stretching mode (near 1600 cm^−1^) in several energetic molecules is on the order of a few picoseconds [31,32]. A recent work using a 2D IR method with the asymmetric NO_2_ stretching mode as the vibrational probe investigated the ultrafast structural dynamics and vibrational energy transfer processes of hexahydro-1,3,5-trinitro-1,3,5-triazine (RDX) in dispersed microcrystals that resembles its bulk phase [33]. These works demonstrated that the 2D IR method possessed both high structural sensitivity and high time resolution, and it could simultaneously excite and probe many vibrational chromophores with varying frequencies [34,35]. Recently, the 2D IR method has attracted extensive attention in the field of the dynamics and structures of interfaces and surfaces as well as condensed-phase molecular systems [36,37,38,39,40,41].

In this work, we theoretically investigate the intermolecular vibration energy transfer processes in TNT/CL-20 using 2D IR spectra. At the same time, the non-covalent intermolecular interaction and the stability of the trigger bond N-NO_2_ are studied using the independent gradient model based on the Hirshfeld partition (IGMH) method [42,43] and the Mayer Bond Order Density (BOD) method [44], respectively. For comparison, the above characteristics of HMX/CL-20 are also investigated. The purpose of this study is to deepen our understanding of the interaction mechanism of cocrystal explosives and provide a theoretical reference for the design of new cocrystal explosives.

## 2. Computational Methods and Theories

Based on the crystal parameters obtained from X-ray crystallographic experimental data [13], the unit cells of TNT/CL-20 and a 3D periodic supercell (2736 atoms, comprising 48 ε-CL-20 molecules and 48 TNT molecules) in 3a×2b×1c arrangement were constructed using Materials Studio 2018 (MS). Cocrystal TNT/CL-20 shows different orientations along with different crystallographic directions; that is, the nitro groups of TNT and the piperazine rings of CL-20 form a repeating zigzagging chain along the [010] direction, and the adjacent CL-20 or TNT molecules form the [001] direction layer structures. Furthermore, interactions occur between the CL-20 nitro groups and the electron-deficient rings of TNT along the [120] direction [13,21]. Therefore, the structure of TNT/CL-20 includes a CL-20 and 3 adjacent TNTs with a total of 99 atom coordinates, as shown in Figure 1 left and Appendix A [16,21]. The molecular structure and atomic coordinates of the cocrystal HMX/CL-20 were taken from its crystal structure determined by X-ray diffraction CCDC (Cambridge Crystallographic Data Center) number 875458 [14], and the asymmetric unit of HMX/CL-20 (Figure 1 right and Appendix A) was constructed according to previous works [18,19]. Geometric optimization and the normal-mode anharmonic vibration frequency calculations of TNT/CL-20 and HMX/CL-20 in the gas phase were performed via density functional theory (DFT) at the level of B3LYP (D3) with the 6-311G(d, p) basis set, for which the relevancy for energetic materials was demonstrated by Osmont et al., as implemented in the Gaussian 16 Program [45,46,47,48,49,50,51]. Based on the Gaussian 16 default convergence criterion, the optimized geometries were characterized as the true localized minimum energy on the potential energy surface without imaginary frequency. Finally, the anharmonic frequencies were assigned as the normal vibration modes through the vibration energy distribution analysis (VEDA) codes [52].

The calculation of quasi-static 2D IR spectra has been described in detail elsewhere [35,38]. Briefly, based on the ground, two-exciton and combination band frequencies of NO_2_ asymmetry stretching modes, two-exciton Hamiltons are
diagonalized, and the anharmonicity, which is essential to creating the 2D IR spectra, is calculated with an anharmonic frequency [50].
The transition dipole moment vectors between the vibrational ground and one-excited state for each of the normal modes are obtained from the DFT
calculation. To obtain the transition dipole moment vector between the
one-exciton to two-exciton states, the harmonic approximation is employed that
is the corresponding ground to one-exciton state transition dipole moment
vector is multiplied by 
$\sqrt{2}$ [35]. In a given vibrational mode, the fourth power
of the involved transition dipole moments is generally proportional to the 2D
IR peak intensity [34,35]. The waiting time T_w_
was set to 3.0 ps.

## 3. Results and Discussions

The experimental and theoretical geometries of CL-20 in TNT/CL-20 are presented in Table 1 [15], while the experimental geometry of HMX/CL-20 is not reported here. We note that the nitrogen atoms in nitro groups are labeled with numbers, and the nitrogen atoms connected with nitro groups in Figure 1 are marked with the corresponding number apostrophe. Since the nitro group has two N-O bonds, the number and the numbers in brackets are used to describe their respective lengths, the same as for the N-C bonds in hexaazaisowurtzitane. The optimized N-N bond length range from 1.392 Å to 1.448 Å, in agreement with the experimental values of 1.383 Å, 1.400 Å and 1.414 Å. Compared with the experimental values of 1.204 Å and 1.215 Å, the theoretical bond lengths of N-O are in a reasonable range of 1.204 Å–1.225 Å. Moreover, the optimized bond lengths of N-C show an excellent agreement with the experimental range of 1.427 Å–1.481 Å, and the theoretical average bond length is 1.456 Å. This is larger than the C=N double bond length of 1.321 Å and close to the CN single bond length, indicating that there is an obvious conjugate interaction between the nitro group and the piperazine ring in CL-20 [13,15]. The conclusion is that the optimized structural parameters of CL-20 in TNT/CL-20 are consistent with the experimental results.

Because the NO_2_ asymmetric stretching modes are usually used as the vibrational probes in energetic materials, the theoretical infrared spectra ranging from 1500 cm^−1^ to 1700 cm^−1^ and the detailed vibrational assignment of NO_2_ asymmetric stretching vibration are presented in Figure 2, Appendix A, respectively. For the spectra of TNT/CL-20 [13,15,16], the lowest transition at ω = 1534.9 cm^−1^ is the NO_2_ group asymmetric stretching vibration and the bending of C-H in TNTIII, which shows an excellent agreement with the experimental value of 1533 cm^−1^ [13,16]. The transition at ω = 1588.6 cm^−1^ is due to the combination of the NO_2_ group asymmetric stretching vibration and the bending of C-H in TNTI and TNTII, as evidenced by the experimental peak at 1588 cm^−1^ [15,16]. The peak at 1596.3 cm^−1^ comes from the NO_2_ group asymmetric stretching vibration and the bending of C-H in TNTII and TNTIII, which is supported by the experimental value of 1597 cm^−1^ [13,15]. The theoretical peaks at 1617.2 cm^−1^ and 1630.4 cm^−1^ are assigned to the NO_2_ group’s asymmetric stretching vibration and the bending of C-H in TNTI and TNTII, respectively, and the corresponding experimental spectra peaks were observed at 1619 cm^−1^ and 1632 cm^−1^ [13,15,16]. The high-frequency peaks (ω = 1655.0 cm^−1^ and ω = 1669.8 cm^−1^) contributed by the NO_2_ asymmetric stretching in CL-20 have no corresponding experimental values. The infrared spectra of HMX/CL-20 have a stronger absorptive intensity than do those of TNT/CL-20. The theoretical peaks at 1559.8 cm^−1^ and 1606.3 cm^−1^, which result from the NO_2_ asymmetric stretching in HMX and CL-20II, agree with the experimental values of 1563 cm^−1^ and 1604 cm^−1^ [14,17,18,19] , respectively. The transition at ω = 1613.7 cm^−1^ is due to the NO_2_ asymmetric stretching in HMX and CL-20I and is consistent with the experimental value of 1615 cm^−1^ [14,17]. Furthermore, the theoretical high-frequency peaks are supported by Mrinal Ghosh’s works [18]. On the basis of the above comparison of the optimized structures and theoretical infrared spectra with the experimental results, the theoretical modes calculated by DFT/B3LYP (D3) with the 6-311G (d, p) basis set can well reproduce the experimental ones in spite of some differences.

As mentioned above, the impact sensitivity of cocrystal energetic materials depends on the rate of mechanical energy transfer, [25,26,27] and the 2D IR spectra have an advantage in detecting the intramolecular and intermolecular vibrational energy transfer due to the unique cross-peak mechanism, that is, the intensity of cross-peaks is proportional to the vibrational energy transfer in the 2D IR spectra [35,36,37,38,39,40,41]. The normalized and simulated 2D IR spectra and the corresponding IR spectra, which show the fundamental frequency and relative intensity of the NO_2_ asymmetric stretching, are presented in Figure 3. The diagonal peaks are a pair of positives (blue in color) and negatives (yellow in color), which represent the ν = 0→1 transition and ν = 1→2 absorptions, [35,36,37,38,39,40] respectively. In the 2D IR spectra of the TNT/CL-20 (Figure 3, left), four positive–negative couplets appear along the diagonal, with their ω_τ_ peak positions corresponding to those in the IR spectrum, and the diagonal peak at ω_m_ = ω_τ_ = 1596.3 overlaps with the diagonal peak at ω_m_ = ω_τ_ = 1588.6. The assignment of these diagonal peaks has been discussed already. Clear 2D IR cross-peaks are shown at (ω_m_, ω_τ_) values of (1617.2, 1588.6) and (1617.2, 1596.3), indicating a complicated vibrational energy transfer process among TNTs, and the cross-peaks in 2D IR spectra are usually caused by the vibrational energy transfer process. The reasons for this are as follows: in a coupled dimer, it is reasonable to consider the one-exciton Hamiltonian for linear spectroscopy [34,35],
(1)$$H=\left(\begin{array}{cc} \hbar \omega_{i} & \beta_{ij}\\ \beta_{ij} & \hbar \omega_{j} \end{array}\right)$$
where ω and *β* represent the frequency and coupling constant between chromophores *i* and *j*, respectively. The exciton eigenvalues are given as follows:(2)$$E_{i,j}=\frac{\hbar \omega_{i}+\hbar \omega_{j}\mp \sqrt{4\beta_{ij}^{2}+{\left(\hbar \omega_{i}-\hbar \omega_{j}\right)}^{2}}}{2}.$$

The eigenstates of *E_i_* and *E_j_* are
(3)$$\left|\phi_{i}\rangle =\cos\alpha \right|10\rangle -\sin\alpha |01\rangle$$
(4)$$\left|\phi_{j}\rangle =\sin\alpha \right|10\rangle -\cos\alpha |01\rangle$$
where α represents the mixing angle
(5)$$\tan2\alpha =\frac{2\beta_{ij}}{\hbar \omega_{j}-\hbar \omega_{i}}.$$

In the weak coupling regime, that is, when the coupling constants are small compared to the frequency splitting,
(6)$$\left|\beta_{ij}\right|\ll \left|\hbar \omega_{i}-\hbar \omega_{j}\right|,$$
their mixing angle α is small, and the exciton states will be localized on the individual sites. The two-excitonic states closely resemble the uncoupled local modes,
(7)$$E_{i,j}\approx \hbar \omega_{i,j}\mp \frac{2\beta_{ij}^{2}}{\hbar \omega_{j}-\hbar \omega_{i}}.$$

Then the coupling constants *β_ij_* can be calculated based on perturbation theory,
(8)$$\beta_{ij}\propto \sqrt{\frac{\Delta_{ij}}{\Delta_{i}}}\left|\omega_{i}-\omega_{j}\right|$$
where ∆_*ij*_ and ∆_*i*_ represent the off-diagonal anharmonicity and diagonal anharmonicity, respectively. 

In the strong coupling regime, that is
(9)$$\left|\beta_{ij}\right|\gg \left|\hbar \omega_{i}-\hbar \omega_{j}\right|,$$
the |*ħω_i_ − ħω*_*j*_| term in Equation (2) is simply ignored to find that the one-exciton eigenstates are perfectly delocalized with α = π/4. In this case, the energies of the excitonic states will split to give
(10)$$E_{i,j}=\frac{\hbar \omega_{i}+\hbar \omega_{j}}{2}\mp \beta_{ij}.$$

The cross-peaks in the 2D IR spectra have a larger separation than the diagonal peaks. Then the coupling constants *β_ij_* can be calculated based on perturbation theory, as follows:(11)$$\beta_{ij}\propto \left|\omega_{i}-\omega_{j}\right|.$$

Based on the Fermi golden rule, the energy transfer rate *κ_ij_* is proportional to the square of the coupling constant *β_ij_;* that is,
(12)$$\kappa_{ij}\propto \beta_{ij}^{2}.$$

To accurately evaluate the off-diagonal and diagonal anharmonicity, the normalized 2D IR spectra between 2 vibrational modes in TNT/CL-20 are shown in Figure 4. The diagonal and off-diagonal anharmonicity values (∆_i_, ∆_ij_) of the peak at ω_m_ = ω_τ_ = 1596.3 cm^−1^ are 47.5 cm^−1^ and 17.5 cm^−1^, which indicates that the coupling regime between vibration modes ω_m_ = ω_τ_ = 1596.3 cm^−1^ and ω_m_= ω_τ_ = 1655 cm^−1^ is a weakly coupled state. Furthermore, based on Equations (8) and (12), the energy transfer rate between CL-20 and TNTII (TNTIII) is proportional to the coefficient 1269.5. The diagonal and off-diagonal anharmonicity values (∆_i_, ∆_ij_) of the peak at ω_m_ = ω_τ_ = 1617.3 cm^−1^ are 3 cm^−1^ and 9 cm^−1^, which shows that the coupling regime between vibration modes ω_m_ = ω_τ_ = 1617.3 cm^−1^ and ω_m_= ω_τ_ = 1655 cm^−1^ belongs to the strong coupling regime. Moreover, according to Equations (9) and (12), the energy transfer rate between CL-20 and TNTI is proportional to the coefficient 1497.7. Therefore, the vibrational energy transfer process between CL-20 and TNTII (TNTIII) is relatively slower than between CL-20 and TNTI.

However, due to the diagonal peaks generated by many NO_2_ asymmetric stretching vibrations in HMX/CL-20, there is no useful information about the energy transfer process between HMX and CL-20s through the 2D IR spectra (Figure 3 right). It can be seen from Equation (8) that the splitting of the cross-peaks is proportional to the strength of the coupling squared in the weak coupling limit. There are several cross-peaks between the vibration modes of ω_m_ = ω_τ_ = 1559.8 cm^−1^ and ω_m_ = ω_τ_ = 1661.0 cm^−1^, indicating that the vibration modes are coupled. However, no cross-peaks appeared between the vibration modes of ω_m_ = ω_τ_ = 1613.7 cm^−1^ and ω_m_ = ω_τ_ = 1661.0 cm^−1^, which shows a weaker coupling effect between them. 

Intermolecular vibration energy transfer is the bridge of intermolecular interaction. It is believed that the intermolecular interaction, including hydrogen bonding, π-stacking, and van der Waals (vdW) forces, is the main contributor to the cocrystal formation [28]. Therefore, we used the IGMH method to analyze intermolecular non-covalent interactions in TNT/CL-20 and HMX/CL-20, where the atomic densities were derived by Hirshfeld partitioning of the actual molecular electron density. [41,42] Figure 5 illustrates the intermolecular non-covalent interaction between TNTs and CL-20 and between HMX and CL-20s. Figure 5 (left) shows that there is mainly a vdW interaction between CL-20 and in TNTs, which has the largest non-covalent interaction isosurface, such as the interaction between N^1^O_2_ and N^12^O_2__._ Due to the conjugation of benzene rings in TNTs, π...π stacking interaction is observed between CL-20 and TNTI, such as the interaction between N^2^O_2_ and N^7^O_2__._ As one might expect, there exists a strong hydrogen bond between the methylene in CL-20 and the N^13^O_2_ group (the black arrow), which has been reported in previous works [13,15,16] and in our 2D IR spectra (Figure 3, left). In HMX/CL-20, the weak interaction is mainly a vdW interaction between HMX and CL-20, which has been previously reported [14,17,18,19]. For example, there are two small green isosurfaces between N^1^O_2_ and N^16^O_2_, and between methylene in HMX and N^5^O_2_ groups, in combination with the π...π stacking interactions. Moreover, the green isosurface between CL-20I and CL-20II is the largest, indicating that the interaction between CL-20s is mainly a vdW interaction, as well, which is confirmed by the 2D IR spectra (Figure 3 right). Through the above comparisons, we find that the non-covalent interactions between TNT and CL-20 are mainly vdW forces and hydrogen bonding, while those between HMX and CL-20 are mainly vdW forces.

Based on the principle of the smallest bond order (PSBO) calculated via quantum chemistry, more sensitive energetic compounds usually contain a smaller bond order of trigger bond in molecules, and it is known that the CL-20 component in TNT/CL-20 decomposes earlier in detonation because CL-20 is more sensitive [24,53]. Therefore, the stability of the N-NO_2_ bond in CL-20 as a trigger bond was investigated using the Mayer BOD approach (Table 2) [44,54]. The Mayer BOD corresponding to the delocalization index in the Hibert atomic space is more comprehensive for investigating the stability of covalent bonds and is defined between *A* and *B* as:(13)$$I_{AB}=I_{AB}^{\alpha }+I_{AB}^{\beta }\;=2\underset{a\in A}{\sum }\underset{b\in B}{\sum }\left[{\left(P^{\alpha }S\right)}_{ab}{\left(P^{\alpha }S\right)}_{ab}+{\left(P^{\beta }S\right)}_{ab}{\left(P^{\beta }S\right)}_{ab}\right]$$
where *P_α_* and *P_β_* are the alpha and beta density matrices, respectively, and *S* is the overlap matrix. The above formula can be equivalently rewritten using the total density matrix P and spin density matrix *P^S^*:(14)$$P=P^{\alpha }+P^{\beta }$$
(15)$$P^{S}=P^{\alpha }-P^{\beta }$$
(16)$$I_{AB}=\underset{a\in A}{\sum }\underset{b\in B}{\sum }\left[{\left(PS\right)}_{ba}{\left(PS\right)}_{ab}+{\left(P^{S}S\right)}_{ba}{\left(P^{S}S\right)}_{ab}\right]$$

For restricted closed-shell circumstances, since the spin density matrix is zero, the formula can be simplified to:
(17)$$I_{AB}=\underset{a\in A}{\sum }\underset{b\in B}{\sum }{\left(PS\right)}_{ba}{\left(PS\right)}_{ab}$$

Generally, the number is positively related to the stability of the same covalent bonds. The Mayer BOD value is close to 1, indicating that the trigger bond of N-NO_2_ is a single covalent bond, which is consistent with the chemical experiments. In TNT/CL-20, it was noticed that the minimum Mayer BOD value is 0.8648 for N6-N, for which the length is the longest (Table 1). Moreover, the length of N2-N is greater than N1-N, while the Mayer BOD value for N2-N is larger than N1-N, indicating that the N2-N bond is more stable. This is because the N^2^O_2_ group has a stronger interaction with TNT (Figure 4). In HMX/CL-20, the Mayer BOD minimum value is 0.8588 and is smaller than that in TNT/CL-20, indicating that HMX/CL-20 is more sensitive than TNT/CL-20. It is interesting that if the NO_2_ group of CL-20 has no intermolecular interaction with TNT or HMX, the Mayer BOD value for N-NO_2_ in CL-20 may be smaller, as shown in N^6^O_2_-N in TNT/CL-20, N^7^O_2_-N in CL-20I, and N^12^O_2_-N in CL-20II (Table 2 and Figure 4). We conclude that the intermolecular interaction can stabilize the trigger bond in TNT/CL-20 and HMX/CL-20, and a stronger intermolecular interaction usually indicates the lower impact sensitivity of the energetic materials. 

## 4. Conclusions

In this work, the intermolecular vibration energy transfer processes and the non-covalent interaction between TNT and CL-20 and between HMX and CL-20 were investigated using 2D IR spectra and the IGMH method by employing DFT/B3LYP (D3)/6-311G (d, p) calculations, respectively. The theoretical geometry and infrared spectra of TNT/CL-20 and HMX/CL-20 are consistent with the experimental results, which indicate that the optimized geometry can well reproduce the experimental geometry, and the theoretical models are reasonable. By comparison, for the off-diagonal anharmonicity and diagonal anharmonicity in 2D IR spectra of TNT/CL-20, where the NO_2_ asymmetric stretching modes were used as the vibrational probe, the intermolecular vibration energy transfer process was calculated in the transition coupling mode. The results show that the coupling regime between CL-20 and TNTII (TNTIII) is a weak coupling regime, but the coupling regime between CL-20 and TNTI is a strong coupling regime, which is supported by the calculated energy transfer rate coefficient. More importantly, the vibration energy transfer process between CL-20 and TNTII (TNTIII) is relatively slower than between CL-20 and TNTI. Because the vibration energy transfer is the bridge of the intermolecular interactions, the weak intermolecular interactions were visualized using the IGMH method, and the results clearly demonstrate that the intermolecular non-covalent interactions include vdW interactions and hydrogen bonds in TNT/CL-20, while it mainly consists of vdW interactions in HMX/CL-20. Furthermore, the Mayer BOD minimum value of the trigger bond in TNT/CL-20 is larger than that in HMX/CL-20, which indicates that the sensitivity of TNT/CL-20 is lower than that of HMX/CL-20. Lastly, the results show that the intermolecular interactions can stabilize the trigger bond in TNT/CL-20 and HMX/CL-20, and stronger intermolecular interactions usually indicate the lower impact sensitivity of energetic materials. 

## Figures and Tables

**Figure 1 molecules-27-02153-f001:**
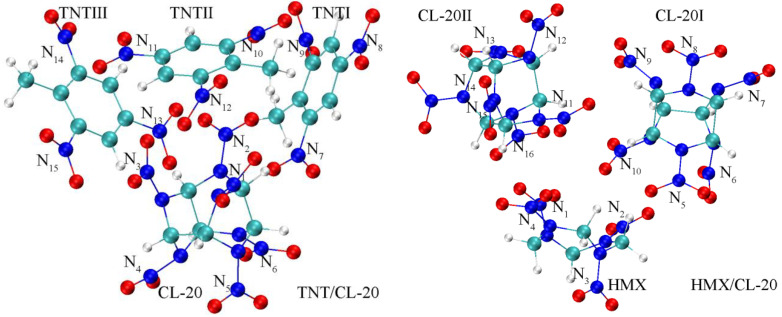
Optimized structures of TNT/CL-20 and HMX/CL-20. Carbon (C), oxygen (O), nitrogen (N), and hydrogen (H) atoms are represented by blue-green-, red-, blue- and white-colored balls, respectively. The nitrogen atoms in nitro groups are labeled by numbers.

**Figure 2 molecules-27-02153-f002:**
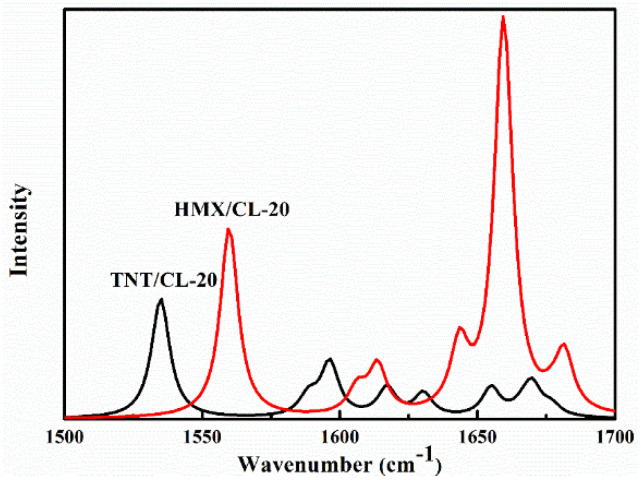
The theoretical infrared spectra of NO_2_ asymmetric stretching in TNT/CL-20 and HMX/CL-20.

**Figure 3 molecules-27-02153-f003:**
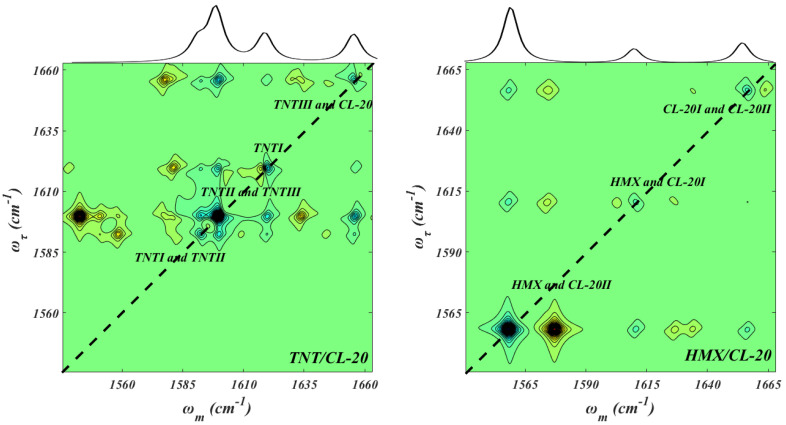
Normalized and simulated 2D IR spectra and corresponding IR spectra (top panel) of NO_2_ asymmetric stretching in TNT/CL-20 (**left**) and HMX/CL-20 (**right**).

**Figure 4 molecules-27-02153-f004:**
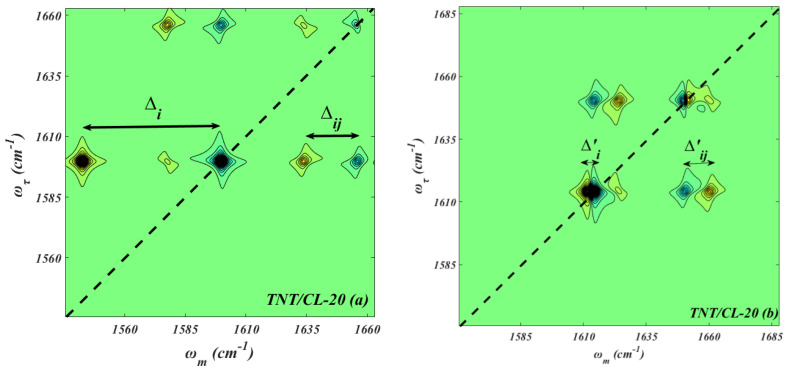
Normalized and simulated 2D IR spectra of NO_2_ asymmetric stretching in TNT/CL-20.

**Figure 5 molecules-27-02153-f005:**
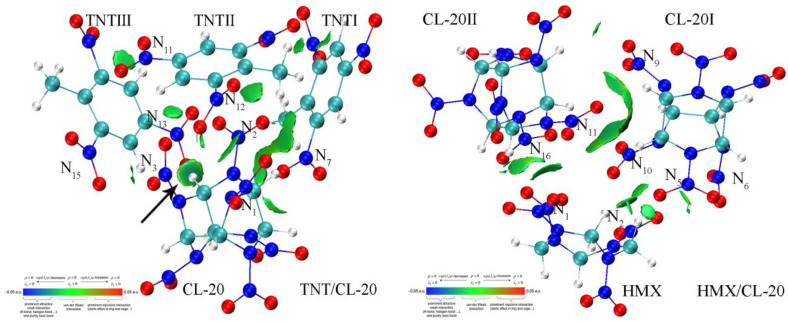
Intermolecular non-covalent interactions between TNT and CL-20 (**left**) and between HMX and CL-20 (**right**). The interactions between CL-20 and TNT (HMX) are presented. Carbon (C), oxygen (O), nitrogen (N), and hydrogen (H) atoms are represented by blue-green-, red-, blue- and white-colored balls, respectively. Only those nitrogen atoms in nitro groups that have interactions with others are numbered.

**Table 1 molecules-27-02153-t001:** Theoretical and experimental bond length (Å) of CL-20 in TNT/CL-20.

Bond Length	Calculation	Experiment
N1-N1’	1.440	
N2-N2’	1.386	1.378
N3-N3’	1.384	1.383
N4-N4’	1.448	
N5-N5’	1.399	1.400
N6-N6’	1.417	1.414
N1-O	1.204 (1.215)	1.204
N2-O	1.214 (1.223)	1.215
N3-O	1.211 (1.225)	
N4-O	1.208 (1.210)	
N5-O	1.213 (1.217)	
N6-O	1.208 (1.212)	
N1’-C	1.454 (1.475)	1.455
N2’-C	1.440 (1.443)	1.445
N3’-C	1.464 (1.466)	1.469
N4’-C	1.451 (1.481)	1.483
N5’-C	1.465 (1.474)	1.478
N6’-C	1.427 (1.433)	1.434

**Table 2 molecules-27-02153-t002:** The Mayer BOD values of N-NO_2_ in CL-20.

TNT/CL-20		HMX/CL-20			
CL-20	Mayer BOD	CL-20I	Mayer BOD	CL-20II	Mayer BOD
N1-N	0.9518	N5-N	0.9255	N11-N	1.0176
N2-N	0.9612	N6-N	0.9238	N12-N	0.8588
N3-N	0.8748	N7-N	0.8718	N13-N	0.9265
N4-N	0.9240	N8-N	0.8821	N14-N	0.8984
N5-N	0.8902	N9-N	0.9120	N15-N	0.8792
N6-N	0.8648	N10-N	0.9198	N16-N	0.8712

## Data Availability

All the data that support the findings of this study are available on request from the corresponding authors.

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
