# Peer review of "Intermolecular Vibration Energy Transfer Process in Two CL-20-Based Cocrystals Theoretically Revealed by Two-Dimensional Infrared Spectra"

_molecules, 2022, doi:10.3390/molecules27072153_

Round 1

Reviewer 1 Report

MOLECULES-1632610: Hai-Chao Ren, Li-Xiang Ji, Tu-Nan Chen, Xian-Zhen Jia, Rui-Peng Liu, Xiu-Qing Zhang, Dong-Qing Wei, Xiao-Feng Wang, and Guang-Fu Ji “Intermolecular Vibration Energy Transfer Process in Two CL-20-based Cocrystals Theoretically Revealed by Two-Dimensional Infrared Spectra”

The authors present their results for their theoretical study in the intermolecular interactions between explosive compounds.  The figures and tables appear to be acceptable for “Molecules”, however, although their material seems to be of the scientific community interest, the different typo mistakes and not adequate wording, distracts too much the reader to follow the topic in a relaxed and understandable manner.  Due to the previously stated I suspended my detailed reading on the manuscript on page 4, and then took a look at the conclusions on page 9.  I will be glad to review the paper again if the wording is edited.  Some additional suggestions are given below.

Suggestions and comments:

  1. Abstract: Rewrite it.

Describe at the start of the abstract that the work was done explosive compounds. TNT and CL-20, HMX.  Also please notice that I suggest also some wording changes to be done on lines 17-20 of the abstract, including the definition of the acronym IGM.

Suggested new text for lines 17-20:

          Inspired by recent co-crystallization and theory of energetic materials, we theoretically investigated the intermolecular vibrational energy transfer process and the non-covalent intermolecular interactions between explosive compounds.  The intermolecular interactions between TNT and CL-20 and HMX and CL were studied using calculated two-dimensional infrared spectra and the independent gradient model method (IGM).

  1. Abstract, line 25.

Specify to what is referred in text as TNTI, TNTII and TNTIII before using their acronyms.

  1. Abstract:

The study of the stability of the trigger bond NO2-N by the Mayer bond order density (BOD) is not mentioned in the abstract.

  1. Page 2, Line 97

Change: “..stability of the trigger bond NO2-N bond are studied…” to “stability of the trigger bond NO2-N are studied…”

  1. Page 3, lines 101 and 102.

Change “explosive” to “explosives.”

  1. Page 4, lines 149 and 153.

Change: “from1.392 Å” to “from 1.392 Å”.

  1. Page 4, line3 and 155.

Change: “~” to “-” in “1.204 Å ~ 1.225 Å” and “1.427 Å ~ 1.481 Å”

  1. Page 4, line 155.

Change: “closed” to “close”.

  1. Page 4, line 155.

Rewrite lines 159-162.

  1. Page 4, line 164

Change: “agree” to “agreement.”

  1. Page 9, line 320

Change: “anahrmonicity” to “anharmonicity.”

Author Response

MOLECULES-1632610: Hai-Chao Ren, Li-Xiang Ji, Tu-Nan Chen, Xian-Zhen Jia, Rui-Peng Liu, Xiu-Qing Zhang, Dong-Qing Wei, Xiao-Feng Wang, and Guang-Fu Ji “Intermolecular Vibration Energy Transfer Process in Two CL-20-based Cocrystals Theoretically Revealed by Two-Dimensional Infrared Spectra”

The authors present their results for their theoretical study in the intermolecular interactions between explosive compounds.  The figures and tables appear to be acceptable for “Molecules”, however, although their material seems to be of the scientific community interest, the different typo mistakes and not adequate wording, distracts too much the reader to follow the topic in a relaxed and understandable manner. Due to the previously stated I suspended my detailed reading on the manuscript on page 4, and then took a look at the conclusions on page 9. I will be glad to review the paper again if the wording is edited.  Some additional suggestions are given below.

Our Response: Your approval of our work has given me lots of encouragement in future work. Thanks for your positive and constructive comments and suggestions. We are very sorry for the mistakes in the manuscript and inconvenience they caused in your reading. The manuscript has been thorough revised and edited by English editing of MDPI, so we hope it can meet the journal’s standard. Thanks so much for your useful comments.

Suggestions and comments:

Abstract: Rewrite it.

Describe at the start of the abstract that the work was done explosive compounds. TNT and CL-20, HMX. Also please notice that I suggest also some wording changes to be done on lines 17-20 of the abstract, including the definition of the acronym IGM.

Suggested new text for lines 17-20:

Inspired by recent co-crystallization and theory of energetic materials, we theoretically investigated the intermolecular vibrational energy transfer process and the non-covalent intermolecular interactions between explosive compounds.  The intermolecular interactions between TNT and CL-20 and HMX and CL were studied using calculated two-dimensional infrared spectra and the independent gradient model method (IGM).

Our Response: We are appreciative of your constructive suggestions. We revised the manuscript based on your constructive suggestions and rechecked the manuscript with the help of English editing of MDPI.

Abstract, line 25.

Specify to what is referred in text as TNTI, TNTII and TNTIII before using their acronyms.

Our Response: Many thanks for your positive and constructive comments and suggestions. We added “2,4,6-trinitrotoluene (TNT) and 2,4,6,8,10,12-hexanitro-2,4,6,8,10,12-hexaazaisowurtzitane (CL-20), 1,3,5,7-tetranitro-1,3,5,7-tetrazocane (HMX)” in the abstract.

Abstract:

The study of the stability of the trigger bond NO2-N by the Mayer bond order density (BOD) is not mentioned in the abstract.

Our Response: Thanks for your helpful suggestion. We add the study of the stability of the trigger bond N-NO2 by the Mayer bond order density (BON) in the abstract.

Page 2, Line 97

Change: “..stability of the trigger bond NO2-N bond are studied…” to “stability of the trigger bond NO2-N are studied…”

Our Response: Thanks for your helpful suggestion. We remove the word “bond” in the abstract and rechecked the manuscript with the help of English editing of MDPI.

Page 3, lines 101 and 102.

Change “explosive” to “explosives.”

Our Response: Thanks for your helpful suggestion. We changed “explosive” to “explosives” on Page 3 , lines 101 and 102.

Page 4, lines 149 and 153.

Change: “from1.392 Å” to “from 1.392 Å”.

Our Response: Thanks for your useful suggestion. We changed “from1.392 Å” to “from 1.392 Å” on Page 4 lines 149 and 153.

Page 4, line3 and 155.

Change: “~” to “-” in “1.204 Å ~ 1.225 Å” and “1.427 Å ~ 1.481 Å”

Our Response: Thanks for your valuable suggestion. We changed “1.204 Å ~ 1.225 Å” and 1.427 Å ~ 1.481 Å” to “1.204 Å - 1.225 Å” and 1.427 Å - 1.481 Å” on Page 4 lines 3 and 155.

Page 4, line 155.

Change: “closed” to “close”.

Our Response: Thanks for your helpful suggestion. We changed “closed” to “close” on Page 4 lines 155.

Page 4, line 155.

Rewrite lines 159-162.

Our Response: Thanks for your valuable suggestion. The sentence was modified “ and the theoretical average bond length is 1.456 Å, which is larger than the C=N double bond length of 1.321 Å and close to the CN single bond length, indicating that there is an obvious conjugate interaction between the nitro group and the piperazine ring in CL-20.” 

 Page 4, line 164

Change: “agree” to “agreement.”

Our Response: Thanks for your helpful suggestion. We changed “agree” to “agreement” on Page 4 lines 164.

 Page 9, line 320

Change: “anahrmonicity” to “anharmonicity.”

Our Response: Thanks for your helpful suggestion. We changed “anahrmonicity” to “anharmonicity” on Page 9 lines 320 and rechecked the manuscript with the help of English editing of MDPI.

Reviewer 2 Report

This paper discusses the intermolecular interaction intermolecular vibrational energy transfer between TNT and CL-20, HMX and CL-20.  Some clarification and attention is needed for describtion of the structures and intramolecular interaction energy. I recommend that this paper be accepted after major revision.

Primary concerns expressed were that:

  • Why these clusters is chosen for DFT calculation? The crystal structure is 1:1 cocrystal and crystal structure form infinite chains along the [001] direction. This need better explanation. Please take a note that direction in the crystallography are described in the square brackets [], whereas the planes are in round brackets ().

  • Figures 2 and 5 presenting the structures are not clear. The color scales and their description on Figure 5 are not readable. I suggest making the carbon atom in different color (gray?) and maybe the balls representing atoms could be smaller.

  • The authors state “Therefore, we use the IGM method to analyze intermolecular non-covalent interaction in TNT/CL-20 and HMX/CL-20 where only molecular geometry is needed”. Of course it need the charge density distribution, Laplacian of charge density and Hessian matrix to perform the analysis of non covalent interaction.

  • The color scale on Figure 5 is showed without values.

  • I recommend adding the atom names to Figure 5, because it is difficult to follow discussion about the intramolecular interactions.

  • The size and number of isosurfaces on Figure 5 depends of course on the selected structural fragments, therefore the reasoning behind selecting of these clusters is crucial.

  • Authors describe the interaction between CL-20 and TNTII as ‘steric effect’. First off do they mean TNTI, and I believe they are simple π...π stacking interactions.

  • Compering both isosurfaces on Figure 5, I cannot agree with the authors statement ‘Through above comparisons, we find that the non-covalent interaction between TNT and CL-20 mainly is vdW force, which between HMX and CL-20 mainly is hydrogen bond.” Where are any hydrogen bond for HXM/CL-20? In both structures green color prevails and yet again adding some molecules to the cluster surrounding HXM would increase the number of non covalent interactions, especially vdW interaction.

Minor concerns were:

- the name NO2-N bond looks odd, maybe N-NO2 would look more natural

Author Response

This paper discusses the intermolecular interaction intermolecular vibrational energy transfer between TNT and CL-20, HMX and CL-20.  Some clarification and attention is needed for describtion of the structures and intramolecular interaction energy. I recommend that this paper be accepted after major revision.

Our Response: Your approval of our work has given me lots of encouragement in future work. Thanks for your positive and constructive comments and suggestions.

Primary concerns expressed were that:

Why these clusters is chosen for DFT calculation? The crystal structure is 1:1 cocrystal and crystal structure form infinite chains along the [001] direction. This need better explanation. Please take a note that direction in the crystallography are described in the square brackets [], whereas the planes are in round brackets ().

Our Response: Many thanks for your positive and constructive comments and suggestions. The cocrystal TNT/CL-20 shows different orientations along different crystallographic directions, that is, the nitro groups of TNT and the piperazine rings of CL-20 form a repeating zigzagging chain along the [010] direction, and the adjacent CL-20 or TNT molecules form the [001] direction layer structures. What’s more, interactions occur between the CL-20 nitro groups and the electron deficient rings of TNT along the [120] direction ( Angew. Chem. Int. Edit. 2011, 123 (38), 9122 and Phys. Chem. Chem. Phys. 2018, 20 (25), 17253). Therefore, we choose the CL-20 and adjacent three TNTs along different directions.

The lattice structure of the cocrystal HMX/CL-20 exhibits molecular packing with one layer of HMX followed by two layers of CL-20, and the asymmetric unit in the manuscript was built based on previous works (Cryst. Growth Des. 2018, 18 (7), 3781 and Nanoscale 2020, 12 (18), 10306).

Moreover, Shi lu et al and Ostrander Joshua S et al conducted the experimental two-dimensional infrared (2D IR)spectra of RDX and PETN, respectively. Then they both used the DFT calculations for theoretical analysis and assigning the NO2 stretching normal modes (J. Phys. Chem. C 2020, 124, 2388 and J. Phys. Chem. B 2017, 121, 1352).

Figures 2 and 5 presenting the structures are not clear. The color scales and their description on Figure 5 are not readable. I suggest making the carbon atom in different color (gray?) and maybe the balls representing atoms could be smaller.

Our Response: Thanks for your helpful suggestion. We replaced the new figures based on your helpful suggestion.

The authors state “Therefore, we use the IGM method to analyze intermolecular non-covalent interaction in TNT/CL-20 and HMX/CL-20 where only molecular geometry is needed”. Of course it need the charge density distribution, Laplacian of charge density and Hessian matrix to perform the analysis of non covalent interaction.

Our Response: Thanks for your constructive suggestion. We changed the “ Independent gradient model (IGM)” to “Independent gradient model based on Hirshfeld partition (IGMH)”. The version of IGM is defined purely based on densities of atoms in their free states, namely it is a method under promolecular approximation (Phys. Chem. Chem. Phys. 2017, 19, 17928), while the version of IGMH is defined based on densities of atoms in Hirshfeld partition, which has a more rigorous physical meaning and can faithfully reflect electronic structure of the system (J. Comput. Chem. 2022, 43, 539).

The color scale on Figure 5 is showed without values.

Our Response: Thanks for your valuable suggestion. We added the values of the color scale on Figure 5.

I recommend adding the atom names to Figure 5, because it is difficult to follow discussion about the intramolecular interactions.

Our Response: Thanks for your helpful suggestion. We added the atom names to Figure 5.

The size and number of isosurfaces on Figure 5 depends of course on the selected structural fragments, therefore the reasoning behind selecting of these clusters is crucial.

Our Response: Thanks for your helpful suggestion. The cocrystal TNT/CL-20 shows different orientations along different crystallographic directions, that is, the nitro groups of TNT and the piperazine rings of CL-20 form a repeating zigzagging chain along the [010] direction, and the adjacent CL-20 or TNT molecules form the [001] direction layer structures. What’s more, interactions occur between the CL-20 nitro groups and the electron deficient rings of TNT along the [120] direction ( Angew. Chem. Int. Edit. 2011, 123 (38), 9122 and Phys. Chem. Chem. Phys. 2018, 20 (25), 17253). Therefore, we choose the CL-20 and adjacent three TNTs along different directions.

The lattice structure of the cocrystal HMX/CL-20 exhibits molecular packing with one layer of HMX followed by two layers of CL-20, and the asymmetric unit in the manuscript was built based on previous works (Cryst. Growth Des. 2018, 18 (7), 3781 and Nanoscale 2020, 12 (18), 10306).

Authors describe the interaction between CL-20 and TNTII as ‘steric effect’. First off do they mean TNTI, and I believe they are simple π...π stacking interactions.

Our Response: Thanks for your constructive suggestion. We revised the mistake and changed the “stetic effect” to “ π...π stacking interactions” using IGMH method in the manuscript.

Compering both isosurfaces on Figure 5, I cannot agree with the authors statement ‘Through above comparisons, we find that the non-covalent interaction between TNT and CL-20 mainly is vdW force, which between HMX and CL-20 mainly is hydrogen bond.” Where are any hydrogen bond for HXM/CL-20? In both structures green color prevails and yet again adding some molecules to the cluster surrounding HXM would increase the number of non covalent interactions, especially vdW interaction.

Our Response: Thanks for your constructive suggestion. We used the IGMH methods and found that the non-covalent interaction between HMX and CL-20 mainly is vdW interaction.

Minor concerns were:

- the name NO2-N bond looks odd, maybe N-NO2 would look more natural

Our Response: Thanks for your valuable suggestion. We changed the name “NO2-N” to “N-NO2” in the manuscript.

Reviewer 3 Report

MOLECULES-1632610: Hai-Chao Ren, Li-Xiang Ji, Tu-Nan Chen, Xian-Zhen Jia, Rui-Peng Liu, Xiu-Qing Zhang, Dong-Qing Wei, Xiao-Feng Wang, and Guang-Fu Ji “Intermolecular Vibration Energy Transfer Process in Two CL-20-based Cocrystals Theoretically Revealed by Two-Dimensional Infrared Spectra”
The authors present their results for their theoretical study in the intermolecular interactions between explosive compounds. The figures and tables appear to be acceptable for “Molecules”, however, although their material seems to be of the scientific community interest, the different typo mistakes and not adequate wording, distracts too much the reader to follow the topic in a relaxed and understandable manner. Due to the previously stated I suspended my detailed reading on the manuscript on page 4, and then took a look at the conclusions on page 9. I will be glad to review the paper again if the wording is edited. Some additional suggestions are given below

Suggestions and comments:
1. Abstract: Rewrite it.
Describe at the start of the abstract that the work was done explosive compounds. TNT and CL-20, HMX. Also please notice that I suggest also some wording changes to be done on lines 17-20 of the abstract, including the definition of the acronym IGM.
Suggested new text for lines 17-20:
Inspired by recent co-crystallization and theory of energetic materials, we theoretically investigated the intermolecular vibrational energy transfer process and the non-covalent intermolecular interactions between explosive compounds. The intermolecular interactions between TNT and CL-20 and HMX and CL were studied using calculated two-dimensional infrared spectra and the independent gradient model method (IGM).

2. Abstract, line 25.
Specify to what is referred in text as TNTI, TNTII and TNTIII before using their acronyms.
3. Abstract:
The study of the stability of the trigger bond NO2-N by the Mayer bond order density (BOD) is not mentioned in the abstract.
4. Page 2, Line 97
Change: “..stability of the trigger bond NO2-N bond are studied…” to “stability of the trigger bond NO2-N are studied…”
5. Page 3, lines 101 and 102.
Change “explosive” to “explosives.”
6. Page 4, lines 149 and 153.
Change: “from1.392 Å” to “from 1.392 Å”.
7. Page 4, line3 and 155.
Change: “~” to “-” in “1.204 Å ~ 1.225 Å” and “1.427 Å ~ 1.481 Å”
8. Page 4, line 155.
Change: “closed” to “close”.
9. Page 4, line 155.
Rewrite lines 159-162.
10. Page 4, line 164
Change: “agree” to “agreement.”
11. Page 9, line 320
Change: “anahrmonicity” to “anharmonicity.”

Author Response

(The authors gave the same response as above.)

Reviewer 4 Report

The manuscript under considearion discusses energy transfer between TNT and hexaazaisowurtzitane. It is very well written, and I really enjoyed reading it. It contains some novelty, although not so much. 

CL-20 abbreviation should be explained when it is first ,emtiomed in the abstract.

Overall, it can be accepted as it is. 

Author Response

The manuscript under considearion discusses energy transfer between TNT and hexaazaisowurtzitane. It is very well written, and I really enjoyed reading it. It contains some novelty, although not so much. 

CL-20 abbreviation should be explained when it is first ,emtiomed in the abstract.

Overall, it can be accepted as it is. 

Our Response: Your approval of our work has given me lots of encouragement in future work. Thanks for your positive and constructive comments and suggestions. We added “2,4,6-trinitrotoluene (TNT) and 2,4,6,8,10,12-hexanitro-2,4,6,8,10,12-hexaazaisowurtzitane (CL-20), 1,3,5,7-tetranitro-1,3,5,7-tetrazocane (HMX)” in the abstract.

Round 2

Reviewer 2 Report

I have no additional comments.